# Exercise Promotes Pro-Apoptotic Ceramide Signaling in a Mouse Melanoma Model

**DOI:** 10.3390/cancers14174306

**Published:** 2022-09-02

**Authors:** Jonghae Lee, Hannah Savage, Shinji Maegawa, Riccardo Ballarò, Sumedha Pareek, Bella Samia Guerrouahen, Vidya Gopalakrishnan, Keri Schadler

**Affiliations:** 1Department of Pediatrics Research, The University of Texas MD Anderson Cancer Center, Houston, TX 77030, USA; 2Graduate School of Biomedical Sciences, The University of Texas MD Anderson Cancer Center, Houston, TX 77030, USA

**Keywords:** physical activity, CerS6, sphingolipid metabolism, cell death, skin cancer

## Abstract

**Simple Summary:**

Exercise has been shown to improve the efficacy of chemotherapy against several tumor models using mice through modulating tumor vascular perfusion, immune function, circulating growth factors, hypoxia, and metabolism in tumor cells and their surrounding microenvironment. However, little is known about the effect of exercise on tumor-cell-intrinsic death mechanisms, such as apoptosis. Ceramide is a bioactive lipid that can promote cell death. The strategy of increasing intracellular ceramide has potential as an anticancer treatment for melanoma with dysregulated ceramide metabolism, but there is not yet a clinically relevant method to do so. We found that moderate aerobic exercise increases pro-apoptotic ceramide in melanoma in mice, and activates p53 signaling, promoting tumor cell apoptosis. This finding suggests that exercise may be most effective as an adjuvant therapy to sensitize cancer cells to anticancer treatments in tumors that exhibit downregulated ceramide generation to evade cell death.

**Abstract:**

Ceramides are essential sphingolipids that mediate cell death and survival. Low ceramide content in melanoma is one mechanism of drug resistance. Thus, increasing the ceramide content in tumor cells is likely to increase their sensitivity to cytotoxic therapy. Aerobic exercise has been shown to modulate ceramide metabolism in healthy tissue, but the relationship between exercise and ceramide in tumors has not been evaluated. Here, we demonstrate that aerobic exercise causes tumor cell apoptosis and accumulation of pro-apoptotic ceramides in B16F10 but not BP melanoma models using mice. B16F10 tumor-bearing mice were treated with two weeks of moderate treadmill exercise, or were control, unexercised mice. A reverse-phase protein array was used to identify canonical p53 apoptotic signaling as a key pathway upregulated by exercise, and we demonstrate increased apoptosis in tumors from exercised mice. Consistent with this finding, pro-apoptotic C16-ceramide, and the ceramide generating enzyme ceramide synthase 6 (CerS6), were higher in B16F10 tumors from exercised mice, while pro-survival sphingosine kinase 1 (Sphk1) was lower. These data suggest that exercise contributes to B16F10 tumor cell death, possibly by modulating ceramide metabolism toward a pro-apoptotic ceramide/sphingosine-1-phosphate balance. However, these results are not consistent in BP tumors, demonstrating that exercise can have different effects on tumors of different patient or mouse origin with the same diagnosis. This work indicates that exercise might be most effective as a therapeutic adjuvant with therapies that kill tumor cells in a ceramide-dependent manner.

## 1. Introduction

The overall survival rate of patients with advanced melanoma is ~50%. This is, in part, because chemotherapy is not effective against advanced melanoma. Accumulating evidence indicates that increasing intracellular ceramide levels enhances the cytotoxic response to chemotherapy of cancer cells [1,2,3,4]. Ceramide is a bioactive lipid molecule that regulates tumor cell proliferation, differentiation, senescence, and apoptosis [5]. Most often, ceramide promotes apoptosis or inhibits proliferation in tumor cells [6]. As a central sphingolipid metabolite, pro-apoptotic ceramide can be used to generate pro-survival signaling by sphingosine-1-phosphate (S1P)- or glucosylceramide-converting enzymes. Stress stimuli trigger the biosynthesis of various ceramide species with specific chain lengths via ceramide synthases in de novo synthesis and salvage pathways, and through acid sphingomyelinase (aSMase)-induced sphingomyelin hydrolysis [7]. Human melanomas exhibit low expression of CerS6, the enzyme that generates pro-apoptotic C16-ceramide [8], and overexpression of acid ceramidase (aCDase), which hydrolyzes ceramide to sphingosine and free fatty acids [9], relative to normal skin tissue.

One potential method to promote pro-apoptotic ceramide signaling in tumors may be through aerobic exercise, though this potential has not been explored. Exercise improves quality of life and physical function, and can reduce the side effects of chemotherapy in cancer patients [10,11]. The present authors and other researchers have demonstrated that exercise improves tumor blood vessel function, leading to increased drug delivery to the cancer cells and improved chemotherapy efficacy against several cancer models in mice [12,13,14,15]. In mice, exercise improves the outcomes of several cancer types by altering tumor cells and the tumor microenvironment, including the tumor immune milieu, hypoxia, circulating growth factors, and cytokines [16,17]. However, tumor-cell-intrinsic changes in response to exercise have not been well characterized. Interestingly, acute exercise has been shown to increase ceramide content in skeletal muscle after acute exercise in healthy adult men [18]. Furthermore, exercise has been shown to increase apoptosis in tumor cells [19,20,21]. Together, the demonstration that exercise can increase ceramide in some tissues, and promote apoptosis in tumors, raises the possibility that exercise increases ceramide accumulation in tumor cells to promote cell death and enhance the response to chemotherapy. The relationship between exercise and ceramide in melanoma has not been evaluated. Given the pro-apoptotic role of ceramide and the pro-survival role of the ceramide metabolism product S1P, it is important to understand the relationship between exercise and ceramide in melanoma cells. 

The primary purpose of this study was to determine the effect of exercise on ceramide metabolism in melanoma in mice. Furthermore, we aimed to determine whether alterations in ceramide metabolism may contribute to the improved efficacy of chemotherapy by exercise. B16F10 and BP melanoma models, which both possess wildtype p53 but have different oncogenic driver mutations, were grown in mice treated with or without treadmill exercise and with or without doxorubicin. We demonstrate that moderate treadmill exercise causes a shift in ceramide metabolism toward pro-apoptotic ceramide accumulation in B16F10 but not BP tumors. Furthermore, exercise increased pro-apoptotic signaling in B16F10. This suggests that exercise may be useful as an apoptosis-promoting factor in combination with standard therapies to treat certain patients with melanoma. These data also indicate that exercise causes different cellular signaling outcomes in different melanomas. 

## 2. Materials and Methods

### 2.1. Cell Culture

B16F10 cells were purchased from American Tissue Culture Collection. BP cells were a kind gift from Dr. Jennifer Wargo (The University of Texas MD Anderson Cancer Center, Houston, TX, USA). B16F10 cells were cultured in RPMI supplemented with 10% fetal bovine serum and 1% penicillin/streptomycin. BP cells were cultured in DMEM containing 10% fetal bovine serum, 1% GlutaMax, and 1% penicillin/streptomycin. Cells were passaged no more than six times, and were harvested by trypsinization at 60–70% confluence on the day of injection. 

### 2.2. Animal Experimental Protocol

Male wildtype C57BL/6J mice were obtained from Jackson Laboratory. All mice were housed under pathogen-free conditions in accordance with the guidelines set by the Institutional Animal Care and Use Committee at the University of Texas MD Anderson Cancer Center. B16F10 tumor cells (1 × 10^5^) or BP tumor cells (8 × 10^5^) in 200 μL of phosphate-buffered saline (PBS) were injected subcutaneously into the flanks of individual mice ranging from 6 to 8 weeks of age. When tumors reached ~30 mm^3^, the mice were assigned to nonintervention control (No Ex) or exercise (Ex) groups. The groups had roughly equivalent average tumor volumes on the first day of the exercise intervention. The treadmill running exercise was performed at 12 m/min for 45 min for 5 consecutive days with 2 days of rest per week for 2 weeks on six-lane mouse treadmills with no incline (Columbus Instruments, Columbus, OH, USA). For experiments using doxorubicin, B16F10 tumor establishment and exercise protocols were the same. Doxorubicin (2.5 mg/kg in 100 µL of PBS) was administered by tail vein twice a week for 2 weeks with a total dose of 10 mg/kg (Appendix A). At the end of the experiment, approximately 48 h after the last exercise treatment, mice were euthanized, tumors were harvested and frozen in OCT, lysed for reverse-phase protein array (RPPA) or Western blot analysis, or sent for liquid chromatography–tandem mass spectrometry (LC/MS) analysis. 

### 2.3. Tumor Volume Assessment

Tumor volume was measured every 2 days using digital calipers. Tumor length and width were recorded to estimate tumor volume using the formula: Tumor volume = Length^2^ × Width × 1/2

Statistical analysis was performed using GraphPad Prism version 9.2.0 for macOS (GraphPad Software, San Diego, CA, USA). Tumor growth was compared between No Ex and Ex mice, and among Con, Ex, Doxo, and Ex + Doxo mice, and analyzed using unpaired *t*-tests and one-way ANOVA, followed by Tukey’s post hoc test. 

### 2.4. Reverse-Phase Protein Array (RPPA)

RPPA was performed on tumors from six mice per group using the Functional Proteomics Reverse-Phase Protein Array Core at MD Anderson Cancer Center. As previously described [22], serial diluted lysates were arrayed on nitrocellulose-coated FAST slide (Whatman) using the Aushon 2470 Arrayer (Aushon Biosystems, Billerica, MA, USA). Each slide was probed with a primary antibody plus a biotin-conjugated secondary antibody. The signal was amplified using the DakoCytomation-catalyzed system (DAKO) and visualized using a 3,3′-diaminobenzidine colorimetric reaction. The slides were scanned, analyzed, and quantified using customized MicroVigene Software (version 5.1.0.0) (VigeneTech Inc., Carlisle, MA, USA) to measure spot intensity. Each dilution curve was fitted with the logistic “Supercurve Fitting” model. Preliminary quality control found that the CF value from one tumor in the exercise group was outside the normal range (normal: 0.25–2.5); thus, that tumor was removed from analysis. Principal component analysis (PCA) and clustering were performed by ArrayTrack Software (version 3.5.0) (Food and Drug Administration, Silver Spring, MD, USA) available at http://edkb.fda.gov/webstart/arraytrack/ (accessed on 28 September 2021). PCA revealed that five of six tumors in each of Con, Doxo, and Ex + Doxo group clustered together, while one in each group was vastly different. The vastly different samples were not included in further analysis so that changes in signaling patterns between groups would be more readily apparent. Functional class annotation analysis was performed on up- or downregulated proteins using the Ingenuity Pathway Analysis (IPA) Software (Ingenuity Systems Inc., Redwood City, CA, USA). 

### 2.5. Lipid Extraction and Sphingolipid Analysis

Tumor tissue sphingolipid masses were quantified using positive mode electrospray ionization/tandem mass spectrometry analysis at the Medical University of South Carolina Lipidomics Core Facility, as described previously [23]. Identifying outliers revealed that one of six tumors in B16F10 Doxo group was a statistical outlier, thus it was removed. Concentrations of ceramide species between two (BP) or four (B16F10) groups were analyzed by either *t*-test or one-way ANOVA, followed by a post hoc analysis, with GraphPad Prism 9.2.0. (GraphPad Software, San Diego, CA, USA).

### 2.6. Immunofluorescence Staining

Tumors frozen in OCT were sectioned at 7 µm. After being fixed in cold acetone for 15 min, slides were washed three times in PBS, blocked with 4% fish gelatin in PBS for 1 h at room temperature, and incubated with primary antibodies diluted in blocking buffer solution overnight at 4 °C. Primary antibodies included ceramide (1:200; C8104; Sigma, St. Louis, MO, USA), CerS6 (1:200; 310550; USBiological, Salem, MA, USA), and cleaved caspase-3 (1:200; 9661; Cell Signaling, Danvers, MA, USA). Alexa Fluor 488 (Thermo Fisher, Waltham, MA, USA) or Alexa Fluor 594 secondary antibodies were used at 1:1000 for 1.5 h at room temperature. Nuclear counterstaining was carried out by Hoechst 33258 or DAPI. Negative control was performed in the absence of these primary antibodies. All images were acquired using a Leica upright fluorescence microscope (DMi8; LEICA Microsystems Inc., Morrisville, NC, USA) with LAS X Software (version 3.7.2.22383) (LEICA Microsystems Inc., Wetzlar, Germany) using the constant camera settings for each protein assessed. Tile scan images of stained sections were obtained by the LAS X Navigator and merged to visualize the entire images of tumor sections. For CerS6 quantification, the pixel-based positive area value of CerS6 against the nucleus above the threshold was obtained by the NIH ImageJ program. To quantify cleaved caspase-3, we determined the number of nuclei using open-source Qupath v0.3.2 (University of Edinburgh, Edinburgh, UK); we counted the positive cleaved caspase-3 signal colocalized with nuclei with the NIH ImageJ plugin Cell counter. For these quantifications, random 10× or 20× magnification images were obtained of each slide, and the individual averages were averaged to determine the group mean and SEM. 

### 2.7. TUNEL Staining

Apoptotic detection was prepared in accordance with the manufacturer’s protocol for the DeadEndTM fluorometric TUNEL system (G3250; Promega, Madison, WI, USA). Images (20× magnification) were processed using a Leica fluorescence microscope system. TUNEL quantification was performed using an identical process for the aforementioned cleaved caspase-3. 

### 2.8. Western Blotting

Tumor homogenates (25 µg) were separated and transferred onto a nitrocellulose membrane. The membrane was blocked with 5% nonfat dry milk or 3% bovine serum albumin in tris-buffered saline with 0.1% Tween 20 (TBST) for 1 h at room temperature. Primary antibodies, including CerS6 (1:500; 310550; USBiological, Salem, MA, USA), aCDase (1:500; 612302; BD Biosciences, Franklin Lakes, NJ, USA), aSMase (1:500; 14609-1-AP; Proteintech, Rosemont, IL, USA), Sphk1 (1:1000; A0660; ABclonal, Woburn, MA, USA), β-actin (1:1000; sc-47778; Santa Cruz Biotechnology, Dallas, TX, USA), and Gapdh (1:1000; 14C10; Cell Signaling, Danvers, MA, USA), were incubated in blocking buffer solution overnight at 4 °C. Horseradish-peroxidase-conjugated secondary antibodies, including anti-mouse (1:2000) and anti-rat (1:2000), were incubated for 1 h at room temperature after being washed three times in TBST. Blots were developed using Pierce ECL reagent (Thermo Scientific; Rockport, IL, USA), scanned with the Chemidoc^TM^ Gel Imaging System (Bio-Rad, Hercules, CA, USA), and analyzed via NIH-ImageJ Software (version 1.53k) (National Institutes of Health (NIH), Bethesda, MD, USA). Internal normalization was performed by comparison of each protein to β-actin and Gapdh. The relative protein values were normalized to those of control No Ex. 

### 2.9. TNM Plot Analysis

The status of ceramide-metabolism-related genes in human cancers was examined using the web platform TNM (Tumor, Normal, and Metastasis) plot (http://tnmplot.com/analysis/) (accessed on 7 January 2022), generating gene expression comparisons from NCBI-GEO, TCGA, TARGET, and GTEx database repositories [24]. *CERS6*, *SMPD1*, *ASAH1*, and *SPHK1* gene levels, as determined by RNA-Seq data, were analyzed in cutaneous melanomas after selecting a platform option, using normal samples from noncancerous patients and further pediatric tissues. A pan-cancer analysis was performed to compare the gene expression patterns in melanoma with those in other cancers. 

### 2.10. Statistical Analysis

Statistical analyses were performed using GraphPad 9.2.0. Unless otherwise noted in corresponding Methods, comparisons between two groups were analyzed by unpaired two-tailed *t*-tests and multiple comparisons were analyzed by one-way ANOVA followed by Tukey’s post hoc test. Data are represented as the mean ± SEM. Differences in *p* values less than 0.05 are considered statistically significant.

## 3. Results

### 3.1. Exercise Induces Apoptosis in B16F10 Melanoma

Exercise improves the efficacy of chemotherapy against tumors in mice; however, the tumor-cell-intrinsic reasons for this improved efficacy remain unclear. To examine the potential mechanisms associated with this improved efficacy, B16F10 tumor-bearing mice were treated with exercise (Ex), doxorubicin (Doxo), or a combination of exercise and Doxo (Ex + Doxo); untreated mice were used as controls (Con). Tumor volumes between Con and Ex mice were not significantly different, and Ex + Doxo was more effective than Doxo alone (Figure 1A), consistent with previous reports [12]. 

RPPA was performed to evaluate changes in protein expression or activation in tumors using 387 antibodies. Principal component analysis (PCA) using 387 proteins showed that tumors from Doxo- or Ex + Doxo-treated mice were similar, while tumors from Ex mice showed a clear separation from all other groups (Figure 1B). These differences were also evident in volcano plots comparing each group; exercise caused the most substantial differences in signaling pathways relative to the control, while the Doxo and Ex + Doxo groups were the most similar to each other (Figure 1C and Appendix A). Hierarchical clustering further demonstrated the unique pattern of expression in tumors from Ex mice relative to all others (Figure 1D). Thus, we focused on comparing signaling within tumors in Ex and Con mice. Relative to the control, exercise increased 70 proteins within tumors, including p53 and cleaved caspase-3. Exercise decreased 53 proteins, including MDM2 (Figure 1C). These changes are consistent with exercise-induced promotion of apoptosis. Interestingly, p53 was upregulated in tumors from mice treated with Ex, either alone or in combination with Doxo (Appendix A). Ingenuity Pathway Analysis (IPA) further indicated that autophagy, the cell cycle, molecular mechanisms of cancer, and apoptosis signaling were the top four upregulated canonical signaling pathways in tumors from Ex relative to Con mice (Figure 1E). Notably, each of the top four signaling pathways upregulated by exercise included p53 as a key signaling protein (Appendix A). Gene pathway analysis by IPA showed that nine proteins that are a part of the canonical apoptosis pathway are significantly different in tumors from Ex relative to Con mice (Appendix A).

### 3.2. Multiple Ceramide Species Are Increased within B16F10 Tumors by Exercise 

As ceramide signaling contributes to the regulation of several of the top signaling changes that are induced by exercise and detected by RPPA, including apoptosis, autophagy, and the cell cycle [25,26,27,28], we determined whether exercise influences ceramide signaling in tumors. LC/MS was used to compare levels of ceramide species between groups. We found a significantly higher concentration of C16 dihydro (dhC16)-ceramide in tumors from Ex relative to Con mice. The most increased ceramide species were dhC16- and C16-ceramides (*p* = 0.040 and *p* = 0.124, respectively, Figure 2A), which are preferentially generated by CerS6. Along with an overall increase in multiple ceramide species induced by exercise, the sum of ceramides was 71.2% higher in tumors from Ex mice than from Con mice (Table 1). In contrast to our expectation, yet in keeping with the RPPA data, there were no increases in ceramide species levels in B16F10 tumors from Doxo or Ex + Doxo mice compared to those in control mice (Table 1 and Figure 2A). 

Given previous reports of the relationship between C16-ceramide and pro-apoptotic signaling [29], and that C16-ceramide directly regulates the expression of p53 [30], we used Pearson’s correlation to determine the relationship between C16-ceramide and p53 levels in B16F10 tumors. There was a trend towards a positive relationship between C16-ceramide and p53 (Figure 2B). Thus, exercise-induced upregulation of p53-associated apoptotic signaling in B16F10 tumors may be related to increased C16-ceramide levels. 

### 3.3. Exercise Increases Apoptosis-Related Ceramide Accumulation and Upregulates CerS6 and Downregulates Sphk1 in B16F10 Melanoma in Mice

We determined the effects of exercise on tumor cell death in the absence of doxorubicin to understand whether exercise may “prime” tumor cells for therapy-induced death. B16F10 tumor-bearing mice were subjected to moderate treadmill exercise (Ex) or no exercise as the control (No Ex). Tumor growth was not different between the two groups, as measured by volume (Figure 3A) or final tumor weight (Figure 3B). To confirm the results of the RPPA indicating increased apoptosis in tumors from Ex mice, cleaved caspase-3 and DNA fragmentation was measured using immunofluorescence and the TUNEL assay. There were significantly more cleaved caspase-3- and TUNEL-positive cells in B16F10 tumors from Ex mice than from No Ex mice (Figure 3C,D). Next, we searched for changes in ceramide and ceramide metabolism enzymes induced by exercise. We found more ceramides in tumor histological sections from Ex mice than in tumors from No Ex mice (Figure 4A), supporting the LC/MS results. Consistent with the accumulation of C16-ceramide, CerS6 was significantly higher in tumors from Ex mice than from No Ex mice, as measured by immunofluorescence (*p* = 0.017, Figure 4B) and Western blot analysis (*p* = 0.006, Figure 4C). There was no significant difference in acid sphingomyelinase (aSMase), acid ceramidase (aCDase), or sphingosine (Sph) levels (Figure 4D,E and Appendix A). However, tumors from Ex mice had roughly half as much Sphk1 as did tumors from No Ex mice (*p* = 0.001, Figure 4F). Together, these findings suggest that exercise increases pro-apoptotic ceramide accumulation in B16F10 tumors potentially through increased CerS6 and decreased Sphk1 (Figure 4G).

### 3.4. Exercise Did Not Increase Ceramide Accumulation in the BP Melanoma Model 

We used a second mouse melanoma model, BP, to determine whether the altered ceramide metabolism observed in B16F10 tumors is broadly representative of the melanoma response to exercise. The BP melanoma model was recently derived from transgenic mice with mutations in Braf and Pten, which are commonly seen in human melanomas (*Braf^V600E/WT^* and *Pten^−/−^*) [31]. Similar to the B16F10 tumor model, tumor growth was not different between No Ex and Ex tumors, as assessed by tumor volume (Figure 5A) and final tumor weight (Figure 5B). Unlike in B16F10 tumors, there were no significant increases in levels of ceramides in BP tumors from Ex mice compared to tumors from No Ex mice (Appendix A and Figure 5C). Unexpectedly, exercise reduced dhC16-ceramide level in BP tumors (*p* = 0.042, Appendix A). Exercise did not change CerS6 and Sphk1 levels in BP tumors, but it increased aSMase and aCDase levels (*p* = 0.009 and *p* = 0.039, respectively, Figure 5D). Despite upregulated aSMase and aCDase, Sph contents did not differ due to exercise in BP tumors (Appendix A). Thus, the regulation of ceramide synthesis induced by exercise may vary not only by cancer type (i.e., melanoma) but also by subtypes within the same disease. The basal levels of ceramides were different between BP and B16F10 tumors. The total ceramide concentration was 30.4% lower in BP tumors than in B16F10 tumors from No Ex mice (1067.9 pmol/mg vs. 1535.0 pmol/mg; Appendix A vs. Table 1).

### 3.5. Human Skin Cancers Have Low CERS6 and High ASAH1 Gene Expression Relative to Normal Skin 

Because of the conflicting findings in the two models of melanoma, we considered which may be more resemblant of melanoma in patients. A gene expression analysis for ceramide-metabolizing enzymes was performed using the web platform TNM (tumor, normal, and metastasis) plot. The mRNA levels of *CERS6* were significantly lower in tumor tissue than in nontumor tissue (Figure 6A). In contrast to *CERS6*, there was more *SMPD1* mRNA, which encodes ceramide-generating acid sphingomyelinase, in skin cancer tissue than in nontumor tissue (Figure 6B). The mRNA levels of *ASAH1* (acid ceramidase, a ceramide-hydrolyzing enzyme), were higher in melanoma than in normal tissue, while *SPHK1* mRNA (encoding sphingosine kinase 1) was not different between groups (Figure 6C,D). To further understand the pattern of *CERS6* gene expression in cancers, we performed a pan-cancer analysis. Surprisingly, lower *CERS6* gene expression was only observed in skin cancer (Figure 6E). Other cancer types with statistical differences between Normal and Tumor groups revealed higher gene expression of *CERS6* (Figure 6E). This skin-cancer-specific pattern of expression was not observed for *SMPD1*, *ASAH1*, or *SPHK1* (Data not shown). Accordingly, low *CERS6* and high *ASAH1* in human cutaneous melanoma may contribute to the low ceramide accumulation in melanoma. 

## 4. Discussion

Identification of the molecular changes within tumors that are induced by exercise is critical to maximizing the potential of exercise as adjuvant therapy for the treatment of cancer. In this study, we evaluated ceramide, a sphingolipid with pro-apoptotic functions, as a potential mediator of exercise-induced tumor cell death. A major finding of this work is that exercise resulted in increased CerS6 protein expression, CerS6-derived dhC16- and C16-ceramides, and p53 pro-apoptotic signaling in B16F10 tumors in mice. These findings suggest that exercise is an important adjuvant for therapy that can increase ceramide-regulated apoptosis. 

The tumor suppressor p53 regulates the expression of genes responsible for growth inhibition, cell senescence, and apoptosis upon DNA damage and oncogenic transformation [32]. Recent studies have demonstrated that CerS6-derived C16-ceramide is crucial for maintaining the stability and enhancing the function of p53 [33,34]. Furthermore, a positive feedback loop of CerS6-derived C16-ceramide-p53 signaling was reported to be an important mechanism to maximize p53 function [35]. Exercise has been shown to activate p53 in some cancer types [36,37]. Here, we speculate that CerS6/C16-ceramide in B16F10 melanoma is interconnected with p53 in response to exercise. RPPA showed increased p53 and decreased MDM2 in Ex relative to No Ex B16F10 tumors, and we found a positive relationship between C16-ceramide and p53 levels. Our findings, together with results in the current literature, suggest that exercise strengthens the positive feedback loop of C16-ceramide and p53 in B16F10 melanoma. 

We found that exercise improved the efficacy of chemotherapy against B16F10 tumors, consistent with the findings of previous reports [12,38]. Melanoma has previously been reported to maintain low ceramide levels by overexpressing ceramide-degrading enzymes and downregulating ceramide-producing enzymes [8,9]. Increasing intracellular ceramide levels is one mechanism by which chemotherapy promotes cell death [2,26,34,39,40]. Our data, demonstrating increased ceramide accumulation in B16F10 tumors from Ex mice, are consistent with the theory that exercise induces ceramide levels to promote chemotherapy-induced cell death. Yet, we did not observe increased C16-ceramide levels in tumors from mice treated with doxorubicin or doxorubicin plus exercise. One potential explanation for this is that the analysis of tumors collected after two weeks of doxorubicin exposure missed the peak in ceramides. The increase in ceramide in tumor cells in response to doxorubicin is likely transient, and ultimately leads to cell death. Thus, by the end of the experiment, those cells that had experienced an increase in ceramide in response to doxorubicin were likely already dead. This needs to be evaluated in a future study. The results of the current study demonstrated that exercise is an effective modality of increasing melanoma ceramide levels, providing a cancer-cell-sensitizing mechanism against chemotherapy. However, further work is warranted to carry out a mechanistic study to determine whether the altered ceramide metabolism in melanoma truly modulates chemotherapy efficacy.

## 5. Conclusions

To our knowledge, this is the first study to compare tumor-cell-intrinsic changes in response to the same exercise intervention in two different models of the same cancer type. As the field of exercise oncology evolves towards personalized exercise prescriptions for each patient, understanding the unique tumor-specific differences in response to the same exercise intervention will be a necessity.

Exercise contributes to tumor cell death. We found that exercise led to increased pro-apoptotic signaling, pushed ceramide metabolism toward a pro-apoptotic ceramide/sphingosine-1-phosphate balance, and resulted in increased tumor cell death in B16F10 melanoma. These findings indicate that exercise is a useful method to promote ceramide-dependent tumor cell death.

## Figures and Tables

**Figure 1 cancers-14-04306-f001:**
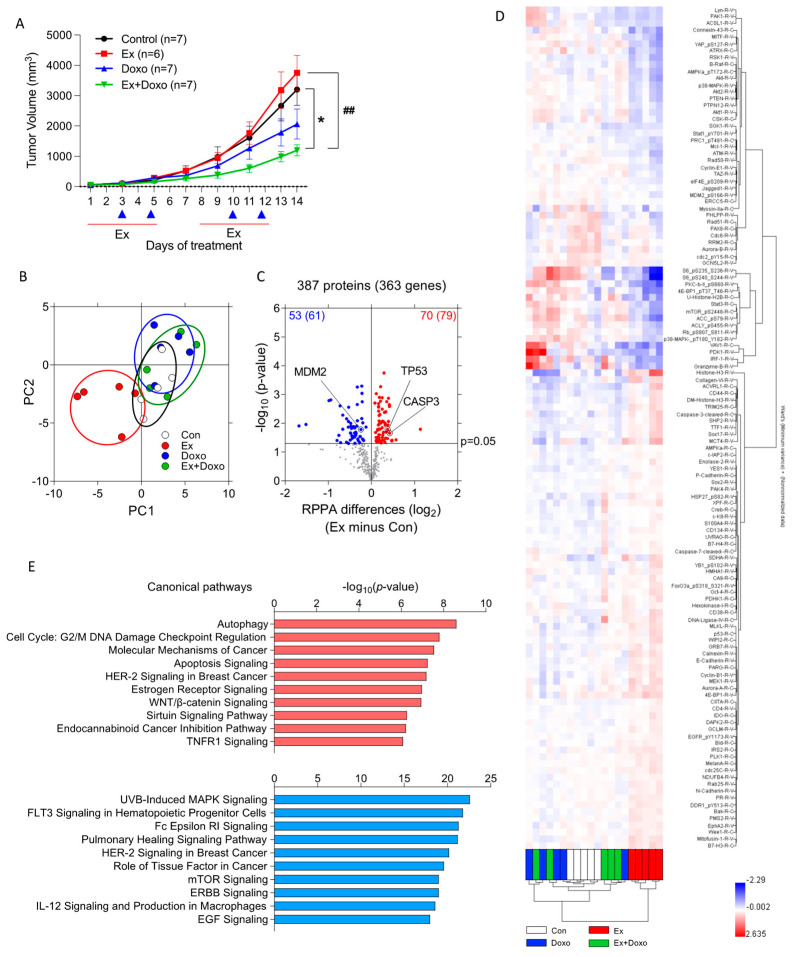
Exercise changes numerous signaling pathways in B16F10 tumors. (**A**) B16F10 tumor volume was measured by caliper over time in mice treated with exercise (Ex), doxorubicin (Doxo), Ex plus Doxo (Ex + Doxo), or no exercise (Con) (*n* = 6 or 7 tumors per group). Red line and blue triangle indicate the day(s) that mice were treatment with exercise and(or) doxorubicin, respectively. Data are shown as mean ± SEM. * *p* < 0.05 vs. Con, ## *p* < 0.01 vs. Ex. (**B**–**D**) RPPA was performed on B16F10 whole-tumor homogenates from Ex, Doxo, Ex + Doxo, or Con mice (*n* = 5 tumors per group). (**B**) Principal component analysis (PCA) summarized different protein signatures from 387 proteins among groups. (**C**) A volcano plot analysis identified significantly increased (red, *n* = 70) and decreased (blue, *n* = 53) proteins in B16F10 tumors from Ex mice compared to tumors from Con mice. The number following the protein numbers indicates the predicted number of changed genes based on RPPA data. The horizontal line indicates *p*-value at 0.05. (**D**) Hierarchical clustering was performed using the 123 significantly changed proteins or protein modifications from the comparison between Ex and Con. (**E**) An Ingenuity Pathway Analysis determined the top 10 upregulated (red) and downregulated (blue) canonical pathways in tumors from Ex relative to Con mice.

**Figure 2 cancers-14-04306-f002:**
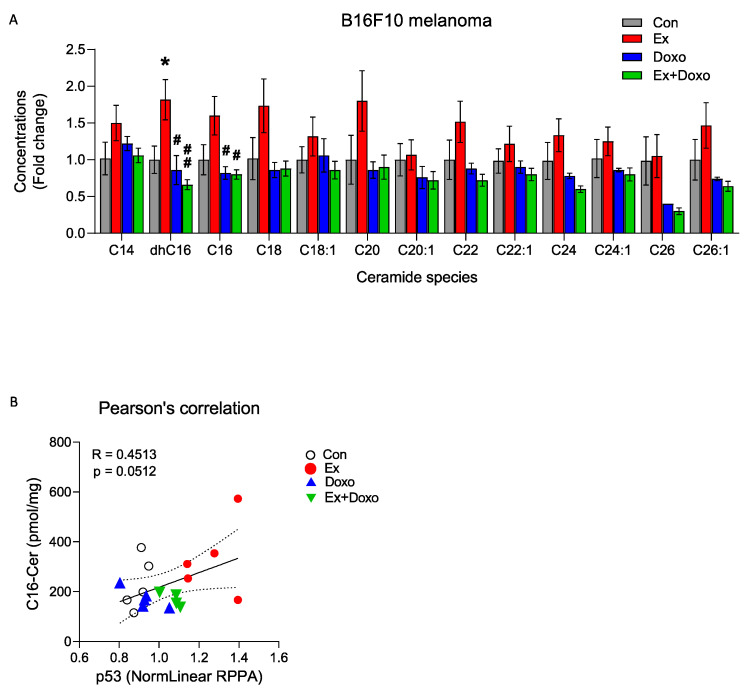
Exercise increases dhC16-ceramide and C16-ceramide levels in B16F10 tumors. (**A**) Liquid chromatography–mass spectrometry (LC/MS) quantification of multiple ceramide species in B16F10 tumors. One-way ANOVA, followed by post hoc tests, was performed to compare differences among groups for each ceramide species (*n* = 5 or 6). (**B**) Pearson’s correlation analysis shows a positive correlation between p53 protein expression and C16-ceramide concentrations determined by RPPA and LC/MS, respectively (*n* = 4–6). Data are shown as mean ± SEM. * *p* < 0.05; # *p* < 0.05; ## *p* < 0.01 vs. Ex.

**Figure 3 cancers-14-04306-f003:**
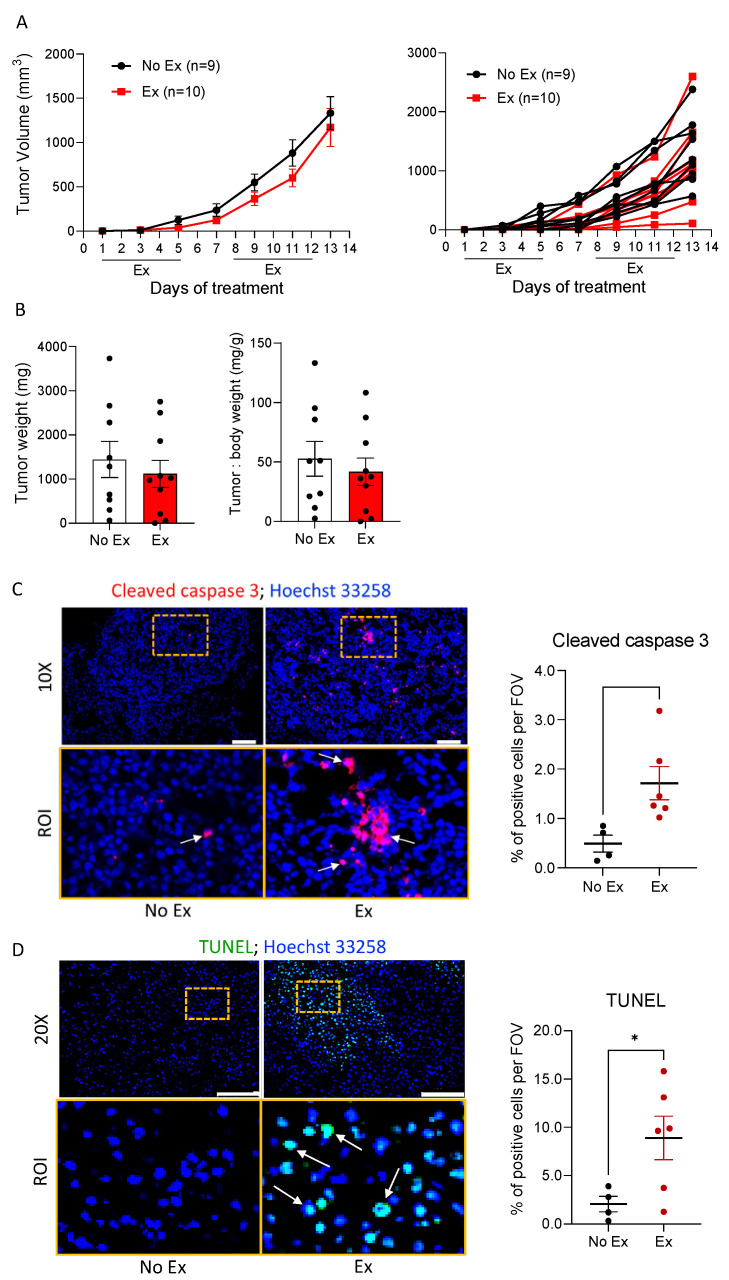
Exercise promotes apoptosis in B16F10 tumors. (**A**) B16F10 tumor volumes were measured by caliper over time in No Ex and Ex mice, and are displayed as the group mean ± SEM and as individual tumors. (**B**) Final tumor weight and the ratio of tumor to body weight were assessed. Black dots indicate an individual value of tumor weight or tumor: body weight. (**C**,**D**) Representative immunofluorescent images of cleaved caspase-3 and TUNEL staining (**upper panels**) were obtained using 10× and 20× objective lenses, respectively. Scale bars = 100 μm. ROI (region of interest) selected from a specific area (dotted yellow box) of 10× or 20× magnification, image-defined patterns of signals to cleaved caspase-3 and DNA fragmentation (**lower panels**). The number of nuclei (Hoechst 33258, blue) and cleaved caspase-3 (red)- or TUNEL (green)-positive cells were counted. Graphs show quantification of the values of co-localized signal on the nucleus in three to eight images of each sample. Black and red dots indicate an individual value in No Ex and Ex, respectively. Data are shown as mean ± SEM. * *p* < 0.05 vs. No Ex.

**Figure 4 cancers-14-04306-f004:**
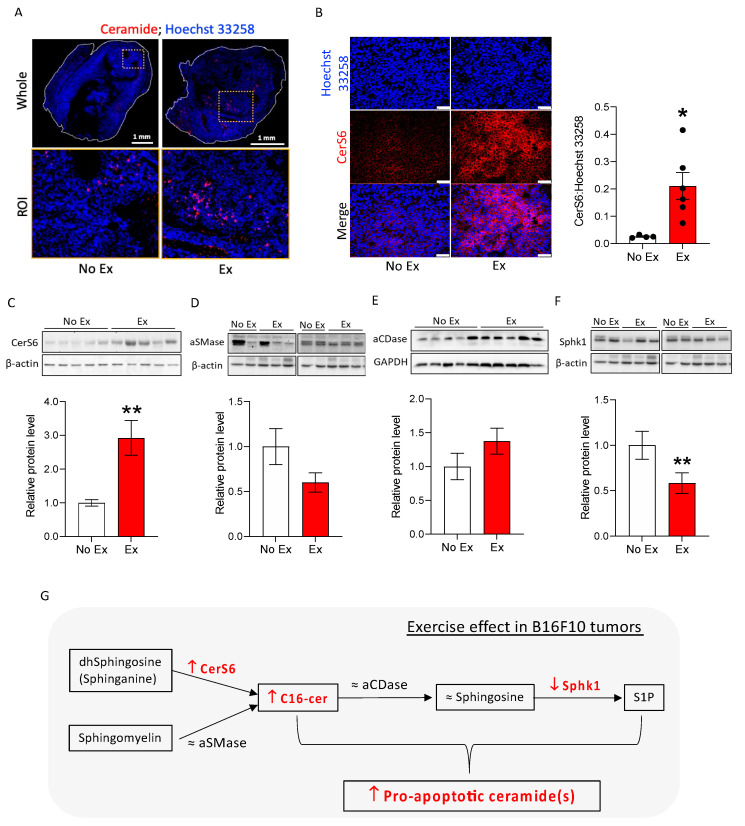
Exercise changes ceramide metabolism enzymes to promote C16-ceramide accumulation in B16F10 tumors. (**A**) Representative immunofluorescent images visualize ceramide (red) and nuclei (blue) in the whole tissue and ROI (region of interest, dotted square in whole) on B16F10 tumor sections. Black dots indicate an individual value of CerS6:Hoechst 33258. (**B**) Representative immunofluorescent images show nuclei (blue) and CerS6 (red). CerS6 was quantified by dividing the area of positive CerS6 staining by the total nucleus area in 2–5 20× images of each sample to obtain one value per tumor (*n* = 2–6). Scale bar = 50 μm. (**C**–**F**) Western blots and quantification graphs show relative protein levels of CerS6 (**C**), aCDase (**D**), aSMase (**E**), Sphk1 (**F**). (**G**) Summary diagram showing that exercise altered ceramide metabolism in B16F10 tumors. Data are shown as mean ± SEM. * *p* < 0.05; ** *p* < 0.01 vs. No Ex. The uncropped blots are shown in Appendix A.

**Figure 5 cancers-14-04306-f005:**
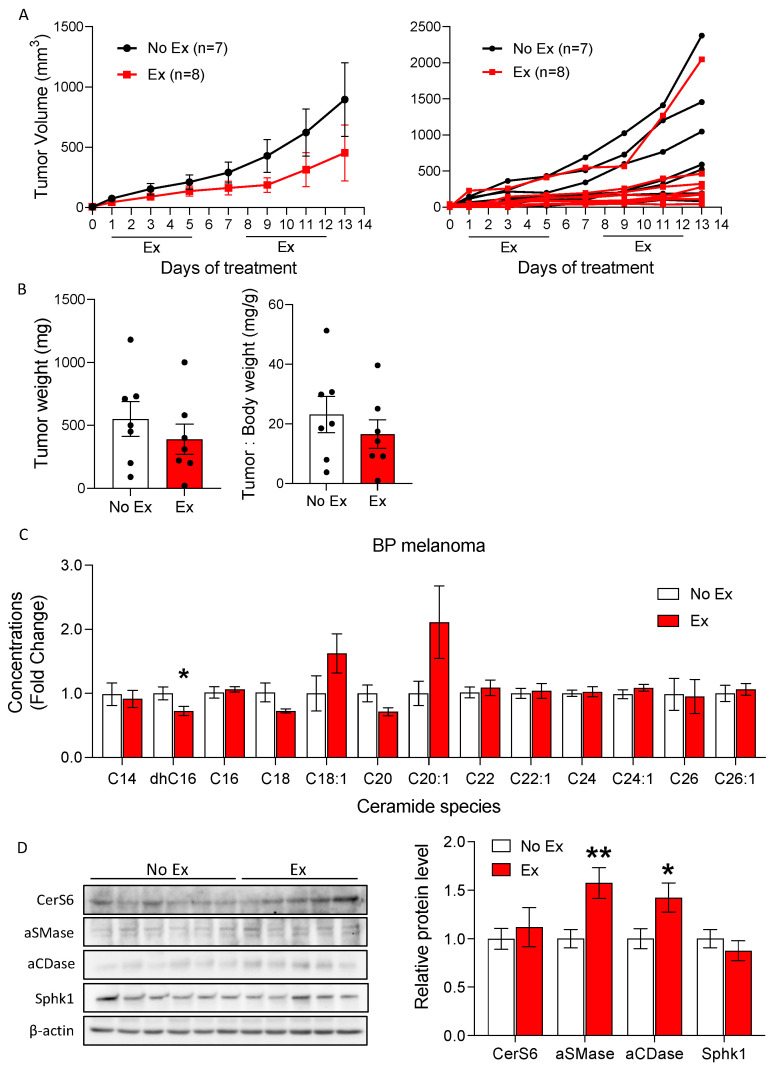
Exercise does not increase ceramide species levels in BP tumors. (**A**) BP tumor volumes were measured by caliper over time in mice treated with or without exercise. Tumor volumes are displayed as the group mean ± SEM and as individual tumors. (**B**) The final tumor weight and ratio of tumor to body weight were assessed. Black dots indicate an individual value of tumor weight or tumor: body weight. (**C**) Liquid chromatography–mass spectrometry (LC/MS) quantification of multiple ceramide species in BP tumors. One-way ANOVA, followed by post hoc tests, was performed to compare intergroup differences between each ceramide species (*n* = 5 or 6). (**D**) Western blot images show CerS6, aCDase, aSMase, Sphk1, and β-actin, and a quantitative bar graph presents the relative protein levels. Data are shown as mean ± SEM. * *p* < 0.05 vs. No Ex; ** *p* < 0.01 vs. No Ex. The uncropped blots are shown in Appendix A.

**Figure 6 cancers-14-04306-f006:**
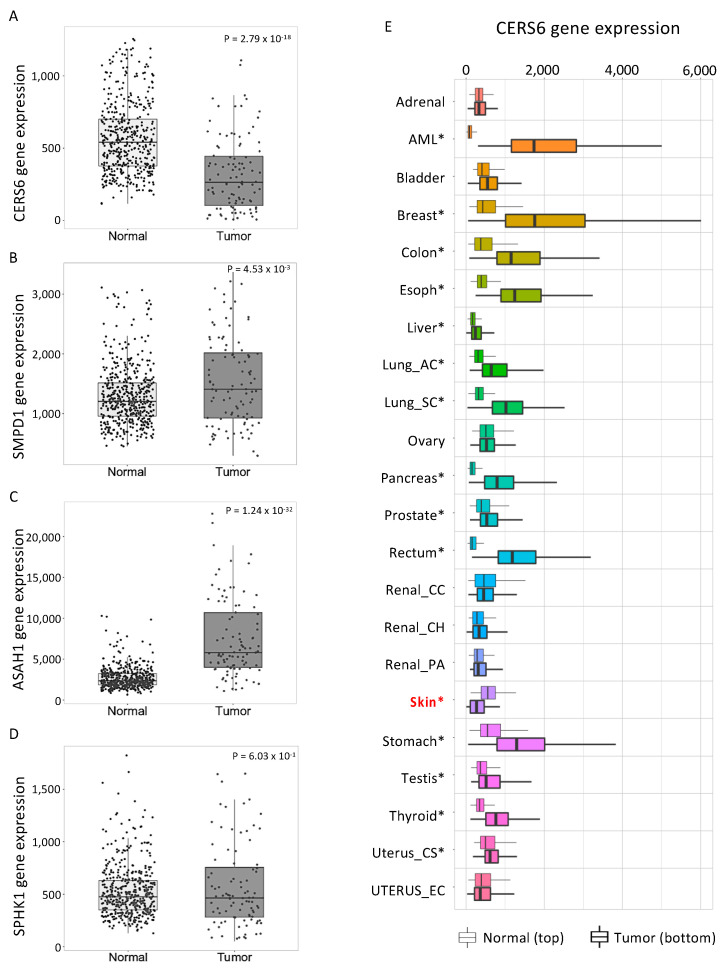
Ceramide metabolism enzyme expression is altered in human skin cancer. The web platform TNMplot was used to evaluate mRNA levels of ceramide metabolism enzymes in human noncancerous normal skin tissue (normal, *n* = 474) and cutaneous melanoma (tumor, *n* = 103). (**A**–**D**) *CERS6* (**A**), *SMPD1* (**B**), *ASAH1* (**C**), and *SPHK1* (**D**) gene levels were analyzed between Normal and Tumor groups. (**E**) Pan-cancer analysis shows *CERS6* gene expressions between Normal (**top**) and Tumor (**bottom**) in 22 cancer tissue types. Lung_AC, lung adenocarcinoma; Lung_SC, lung squamous cell carcinoma; Renal_CC, kidney renal clear cell carcinoma; Renal_CH, kidney chromophobe; Renal_PA, kidney renal papillary cell carcinoma; Uterus_CS, uterine carcinosarcoma; Uterus_EC, uterine corpus endometrial carcinoma. Significant differences were analyzed by Mann–Whitney U test and are denoted with an asterisk. * *p* < 0.01 vs. paired normal control. Font and color were modified for better readability of the original graphs obtained from TNM plot.com. Skin cancer is marked with red for better visualization.

**Table 1 cancers-14-04306-t001:** Absolute concentration of ceramide species (pmol/mg) homogenates from B16F10 melanoma from Con, Ex, Doxo, and Ex + Doxo mice.

Ceramide	Con (*n* = 6)	Ex (*n* = 6)	Doxo (*n* = 5)	Ex + Doxo (*n* = 5)
C14	5.7 ± 1.3	8.6 ± 1.4	6.9 ± 0.5	6.1 ± 0.6
dhC16	24.5 ± 4.6	44.7 ± 6.7 *	21.4 ± 5.0 #	16.3 ± 1.8 ##
C16	213.0 ± 43.1	336.0 ± 55.7	172.8 ± 17.9 #	171.9 ± 11.2 #
C18	62.5 ± 18.0	108.6 ± 23.1	54.4 ± 5.6 #	53.7 ± 6.3 #
C18:1	3.8 ± 0.7	5.0 ± 1.0	4.0 ± 0.9	3.2 ± 0.5
C20	91.6 ± 29.5	165.0 ± 37.2	77.6 ± 10.9	82.0 ± 14.9
C20:1	1.9 ± 0.4	2.0 ± 0.4	1.4 ± 0.3	1.4 ± 0.2
C22	262.0 ± 71.1	390.2 ± 72.2	229.9 ± 21.6	197.1 ± 20.3
C22:1	24.7 ± 4.3	29.5 ± 6.0	22.5 ± 2.0	19.8 ± 1.8
C24	633.3 ± 161.8	841.7 ± 142.0	481.2 ± 24.0	384.7 ± 38.2
C24:1	456.7 ± 118.1	571.6 ± 86.5	396.3 ± 17.5	365.7 ± 43.0
C26	16.0 ± 5.2	17.0 ± 4.6	6.4 ± 0.2	4.6 ± 0.5
C26:1	8.7 ± 2.4	12.7 ± 2.6	6.4 ± 0.2	5.6 ± 0.6
Sum	1804.3 ± 441.4	2532.5 ± 431.4	1481.0 ± 58.5	1312.1 ± 130.3

* *p* < 0.05 vs. Con, # *p* < 0.05; ## *p* < 0.01 vs. Ex.

## Data Availability

The data presented in this study are available in the Appendix A.

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
