# Peer review of "Exercise Promotes Pro-Apoptotic Ceramide Signaling in a Mouse Melanoma Model"

_cancers, 2022, doi:10.3390/cancers14174306_

Round 1

Reviewer 1 Report

The manuscript titled “Exercise promotes pro-apoptotic ceramide signaling in a mouse melanoma model” by Lee et. al. is talking about the effect of exercise on the progression of cancer via ceramide signaling in the mice model. The authors have shown that aerobic exercise increases pro-apoptotic ceramide promoting tumor cell apoptosis in melanoma. This finding suggests that exercise may be most effective as an adjuvant therapy to sensitize cancer cells to anti-cancer treatments in tumors that have ceramide dependent. The manuscript is well written and well organized, and the concept was proved with proper experiments. A few minor comments are as follows.

1.      The connection between the exercise and ceramide accumulation inducing apoptosis is poorly described and the reference provided (ref no 17) does not explain about the apoptosis, how it was hypothesized “increases ceramide accumulation in tumor cells to promote cell death and enhance the response to chemotherapy”?

2.      What is BP melanoma, which appears first time in line, 71, it would be better to elaborate on the BP

3.      Why two types of cancer cells B16F10 tumor cells or BP tumor cells were used, what was the objective to use these two types of cancer cells?

4.      B16F10 tumor cells or BP tumor cells injected subcutaneously in separate animals or in the same animals? Line 93 and 94.

5.      Why male mice were used only and what was the age and weight of the mice used for the experiments?

6.      How many animals (n =?)  were used per experiment?

7.      Did tumors were tested to confirm the melanoma by using melanoma-specific markers before doing the downstream experiments?

8.      What is the scale bar for all the IHC images?

9.      Why there are multiple bands in the western blot images in No Ex and Ex groups, in Fig -3, and so on, does it is from all the animals used for the experiments?

Reviewer 2 Report

This study explored the “Exercise promotes pro-apoptotic ceramide signaling in a mouse melanoma model”. This study is very interesting with well-written. It is the first study to compare tumor cell-intrinsic changes in response to the same exercise intervention in two different models of the same cancer type. The findings suggest that exercise is a usual method to promote ceramide-dependent tumor cell death. But there are some problems here that need to be solved.

1. The keywords should not include the No. behind each word.

2. When the rats run on the treadmill, does the treadmill have a ramp angle?

3. For 12 meters/minute for 45 minutes running, what is the intensity at this speed? How does it apply to humans in the future?  

Author Response

Please see the attachment, "Response to reviewer #2".
